# A sequential methodology for the rapid identification and characterization of breast cancer-associated functional SNPs

Yihan Zhao[1,2], Di Wu [3,4], Danli Jiang[1], Xiaoyu Zhang [1], Ting Wu[1,5], Jing Cui[6], Min Qian[2], Jean Zhao[7], Steffi Oesterreich [8,9], Wei Sun[10], Toren Finkel[1,10] & Gang Li [1,10✉]

GWAS cannot identify functional SNPs (fSNP) from disease-associated SNPs in linkage disequilibrium (LD). Here, we report developing three sequential methodologies including Reel-seq (Regulatory element-sequencing) to identify fSNPs in a high-throughput fashion, SDCP-MS (SNP-specific DNA competition pulldown-mass spectrometry) to identify fSNP-bound proteins and AIDP-Wb (allele-imbalanced DNA pulldown-Western blot) to detect allele-specific protein:fSNP binding. We first apply Reel-seq to screen a library containing 4316 breast cancer-associated SNPs and identify 521 candidate fSNPs. As proof of principle, we verify candidate fSNPs on three well-characterized loci: *FGFR2*, *MAP3K1* and *BABAM1*. Next, using SDCP-MS and AIDP-Wb, we rapidly identify multiple regulatory factors that specifically bind in an allele-imbalanced manner to the fSNPs on the *FGFR2* locus. We finally demonstrate that the factors identified by SDCP-MS can regulate risk gene expression. These data suggest that the sequential application of Reel-seq, SDCP-MS, and AIDP-Wb can greatly help to translate large sets of GWAS data into biologically relevant information.

[1] Aging Institute, University of Pittsburgh, Pittsburgh, PA 15219, USA. [2] School of Life Sciences, East China Normal University, Shanghai, China. [3] Adams School of Dentistry, University of North Carolina at Chapel Hill, Chapel Hill, NC 27599, USA. [4] Department of Biostatistics, University of North Carolina at Chapel Hill, Chapel Hill, NC 27599, USA. [5] Department of Medicine, Xiangya School of Medicine, Central South University, Changsha, China. [6] Department of Medicine, Brigham and Women's Hospital, Boston, MA 02115, USA. [7] Department of Chemical Biology, DFCI, Boston, MA 02115, USA. [8] Department of Pharmacology & Chemical Biology, University of Pittsburgh School of Medicine, Pittsburgh, PA 15261, USA. [9] Women's Cancer Research Center, Magee-Women's Research Institute, University of Pittsburgh Cancer Institute, 204 Craft Avenue, Pittsburgh, PA 15213, USA. [10] Department of Medicine, Division of Cardiology, University of Pittsburgh Medical Center, Pittsburgh, PA 15219, USA. ✉email: lig@pitt.edu

Breast cancer (BC) is the most common cancer in women both in the developed and less developed world, accounting for 11.6% of all new cancer cases and 6.6% of the total cancer deaths in 2018 [1]. BC is strongly correlated to age, and for US women, the median age of diagnosis is 62 (SEER Cancer Statistics Review, http://seer.cancer.gov/csr/1975_2015/). Based on available molecular and genetic information, BC can be divided into five main intrinsic or molecular subtypes. These include luminal A, luminal B, triple-negative/basal-like breast cancer (TNBC), HER2-enriched and normal-like BC[2]. Among these, TNBC is the most aggressive form of the disease.

BC risk is strongly influenced by rare coding variants in susceptibility genes such as *BRCA1* and *BRCA2*, although these variants account for only 5−10% of BC cases in the general population[3,4]. In addition, many common genetic variants, mainly in the form of single nucleotide polymorphism (SNP) in the noncoding region of human genome, also contribute to the disease susceptibility. In general, although these more common variants each confers small individual risk, combinations of these SNPs, together with environmental factors, are believed to contribute significantly to the etiology of the disease[5–7]. However, the mechanisms underlying these genetic contributions remain incompletely characterized.

Since 2005, genome-wide association studies (GWAS) have identified more than 170 genetic loci that are associated with BC[8,9]. However, in general, translating GWAS data into biological insights has remained challenging. There are a number of reasons for this, but one major hurdle is that SNPs used by GWAS to pinpoint genetic loci are simply a proxy for large stretches of DNA (haplotypes) that contains many other SNPs in linkage disequilibrium (LD), most of which reside in noncoding DNA. Therefore, identifying a functional SNP (fSNP) as a causal factor for a disease among all the other disease-associated SNPs in LD is often difficult. While in silico methods exist, these strategies are by nature computational, and therefore often lack experimental validation. Even where a fSNP is established, it often remains quite difficult to define the mechanisms by which a fSNP regulates the expression of its disease-associated target gene. Indeed, while several thousand GWAS have combined to identify thousands of disease-associated loci as bona fide disease risk factors, there are relatively few studies that have identified and completely characterized fSNPs[10,11].

To address this question, previously we developed an unbiased high-throughput (HTP) technique we termed SNP-seq[12]. In this study, we introduce another unbiased HTP technique referred to as Reel-seq (regulatory element-sequencing) to identify fSNPs in an even more efficient way. We also herein describe two additional tools we call SNP-specific DNA competition pulldown-mass spectrometry (SDCP-MS) and allele-imbalanced DNA pulldown-Western blot (AIDP-Wb). These assays are developed because of the general belief that noncoding fSNPs influence disease susceptibility by binding regulatory factors involved in gene expression, and do so, in an allele-specific fashion. As such, these new techniques provide a convenient methodology to identify the regulatory proteins that bind to fSNPs and ascertain allele-specific protein: fSNP binding. We apply these techniques to rapidly identify 521 candidate fSNPs from a library containing 4316 BC-associated SNPs located on 177 risk loci in LD with $R^2 > 0.8$. Furthermore, by coupling Reel-seq with SDCP-MS and AIDP-Wb, we provide a strategy to delineate which regulatory proteins bind to a fSNP, and demonstrate how this binding is allele-specific and examine how these regulatory proteins alter risk gene expression.

## Results

**Reel-seq.** To identify noncoding fSNPs, we developed Reel-seq as described in Fig. 1. Reel-seq is designed to identify fSNPs based on the fact that the majority of disease-associated SNPs are located in noncoding regions of human genome[7]. The regulatory elements containing these SNPs presumably exert their function by binding to specific regulatory proteins that modulate risk gene expression[13–16]. In brief, Reel-seq is an electrophoresis mobility shift assay (EMSA)-based, unbiased HTP screening technique. For screening, a SNP sequence library is generated by massive parallel oligonucleotide synthesis containing disease-associated SNP constructs with both the risk and non-risk alleles. In each of the constructs, a 31 bp SNP centered sequence is placed between two primers for PCR amplification, as well as for next-generation sequencing (NGS) (Fig. 1a). Since a typical transcription factor occupies 6−12 nucleotides[17], a 31 bp fragment will, in general, be enough to span the entire binding motif for a given transcription factor. To perform the screen, libraries are mixed with/without nuclear extract (NE) isolated from a disease-relevant cell type and then analyzed on a TBE native polyacrylamide gel for gel shifting (Fig. 1b, left). After electrophoresis, the unshifted libraries in both buffer-treated controls and NE-treated samples are isolated and amplified by PCR. The amplified libraries are then used for another round of screening using the same gel shifting procedure, for a total of ten cycles. If a SNP is functional, it will bind to a regulatory protein(s) with the two alleles shifted differentially based on their binding affinity to the protein(s) as illustrated by SNP2 in Fig. 1b. Over several cycles, the ratio of the two alleles will be altered in the unshifted libraries and this alteration will be increasingly enriched after each cycle of gel shifting when one compares the NE-treated samples with the buffer-treated controls (Fig. 1b, right). If the SNP is not functional, then the ratio of the two alleles will not be altered either because there are no proteins bound to the SNP or because each allele binds regulatory protein(s) with equal affinity (e.g. SNP1 in Fig. 1b).

**Identification of 521 candidate fSNPs associated with BC.** To demonstrate the feasibility of Reel-seq, we initially screened a synthetic DNA library that contains 4316 SNPs in LD with $R^2 > 0.8$ located on 177 BC-associated loci based on GWAS Catalog (2015) as outlined in Fig. 2a. In total, we screened 8676 SNP sequences representing both risk and non-risk alleles, including 30 SNPs with three alleles and seven SNPs with four alleles. We generated this library through massive parallel oligonucleotide synthesis (Mycroarray). As a proof of principle, we used NE isolated from TNBC cell line MDA-MB-468 as the source of regulatory proteins to screen this library. MDA-MB-468 cells were purchased from ATCC (Cat#: ATCCHTB-132) and were free of mycoplasma. For screening, we performed each run of the gel shift assay with five buffer-treated controls and five NE-treated samples. In each assay, 50 ng libraries were incubated with/without 10 μg NE. After separating DNA−protein complexes using a 6% TBE native polyacrylamide gel, the unshifted DNA libraries from five controls and five samples were purified and amplified by PCR for the next round of gel shifting analysis. After a total of ten cycles, 40 libraries from cycles 1, 4, 7, and 10 (five buffer-treated controls and five NE-treated samples for each cycle) were prepared for Nextseq by incorporating 40 barcodes according to standard NGS Illumina protocol[18].

After sequencing, we obtained $4.82 \times 10^8$ reads, among which we identified $1.91 \times 10^8$ reads with a perfect sequence match to the library template with a rate of 40%. Quality control was then performed to eliminate all the SNPs that did not have a completed set of reads on the two alleles from cycles 1 to 4, 7, and 10. By doing so, we recovered 6785 sequences representing a total of 3436 SNPs (Source data are provided as a Source Data file). The quality of this screen was evidenced by high reproducibility among the five repeats with all correlation

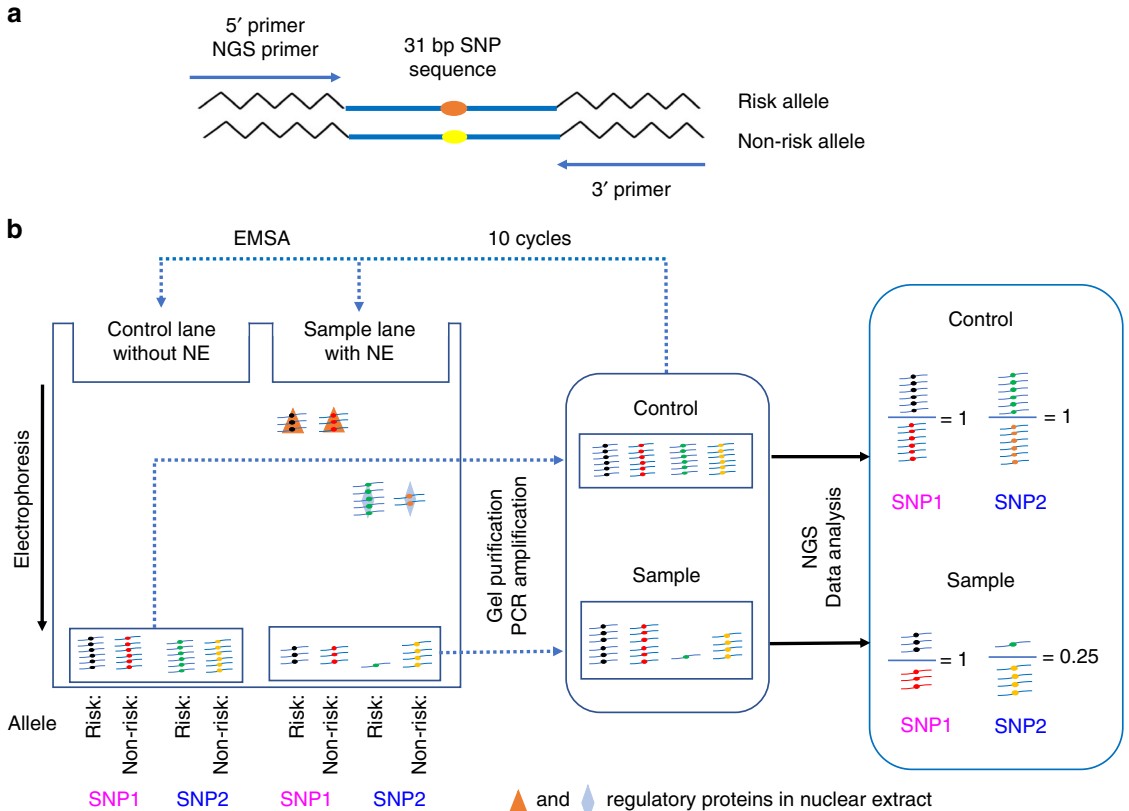

**Fig. 1 Reel-seq for fSNP identification. a** Reel-seq library construct. **b** Reel-seq to identify fSNPs. SNP1: a non-fSNP exhibiting a balanced gel shift pattern between risk and non-risk allele (left in purple); therefore, showing no difference in the ratio of risk and non-risk allele (right in purple) between the buffer-treated control and the NE-treated sample. SNP2: fSNP demonstrating allele-imbalanced gel shift pattern between the risk and non-risk allele (left in blue); therefore, showing significant difference between the buffer-treated control and the NE-treated sample (right in blue). Nonspecific shifting is not shown on EMSA, but it is considered to be evenly and randomly distributed. EMSA electrophoresis mobility shift assay, NE nuclear extract, NGS next-generation sequencing.

coefficient demonstrating an $R^2 > 0.8$ (Fig. 2b) (Source data are provided as a Source Data file). To identify fSNPs, we first calculated the ratio of the risk allele versus the non-risk allele from the five buffer-treated controls and the five NE-treated samples for each SNP at cycle 10. We then calculated the $P$ value using a Student's $t$ test to determine whether there was a significant difference by comparing the ratios from the five controls with that from the five samples for each SNP. Using this method, we identified 1719 SNPs exhibiting a significant difference ($P$ value < 0.05) between the ratio of the five buffer controls and the five NE-containing samples measured at cycle 10. Subsequently, we applied a second filter to these 1719 SNPs by determining whether the ratio between the five controls and the five samples progressively increased across cycles 1, 4, 7 and 10 with an empirical $Slope > 0.05$ or $< -0.05$ as would be predicted if a SNP is functional (Fig. 2c). The rationale to this $Slope$ cutoff is shown in Fig. 2c. Using this strategy, we identified a subset of 521 SNPs with $P$ value < 0.05 and $Slope > 0.05$ or $< -0.05$ and assigned these SNPs as candidate fSNPs, while we denoted the remaining 1198 SNPs with $P$ value < 0.05, but $Slope < 0.05$ or $> -0.05$ as putative fSNPs (Fig. 2d).

In general, in Reel-seq screening, we want to keep as many positives as possible; therefore, we did not use any multiple testing adjustment for the $P$ value. In this case, we are aware of the probability of excessive false positives at the end of our data analysis using the Reel-seq screen. However, later downstream validation steps such as allele-imbalanced gel shifting and luciferase reporter assays were used to narrow this initial pipeline.

In an attempt to demonstrate that our 521 candidate fSNPs identified by Reel-seq were indeed functional, we decided to more closely analyze the fSNPs on the *FGFR2*, *MAP3K1*, and *BABAM1* loci. These three loci were chosen because of their presumed biological relevance since *FGFR2* and *MAP3K1* are both among those loci known to demonstrate a strong association with BC[19]. *FGFR2* acts through multiple downstream signaling pathways such as *RAS-MAPK*, *PLCγ*, *PI3K*, and *JAK/STAT* that play vital roles in cell proliferation, survival, differentiation, and drug resistance. Mutations on *FGFR2* have been identified in both ER+ and ER- BCs[20]. MAP3K1 is a serine/threonine kinase and is part of multiple signal transduction cascades, including the ERK and JNK kinase pathways, as well as the NF-kappa B pathway. Recent large-scale genomic studies have revealed that *MAP3K1* copy number loss and somatic missense or nonsense mutations are observed in a significant number of different cancers[21]. BABAM1 has been identified as playing an important role in DNA damage repair and checkpoint control by maintaining the integrity and stability of BRCA1-A complex[22]. In total, Reel-seq identified five candidate fSNPs from *FGFR2* locus (rs7895676, rs2981578, rs2981584, rs4752570, and rs1219642), five from the *MAP3K1* locus (rs16886034, rs60054381, rs74762363, rs77371588, and rs111968853) and two from the *BABAM1* locus (rs79321361 and rs8101691). Consistent with the designation of these 12 SNPs as candidate fSNPs, we could demonstrate that all these 12 SNPs exhibited allele-imbalanced gel shifting using NE from MDA-MB-468 cells and all the shifted allele-imbalanced bands could be specifically competed away with an increased amount of the

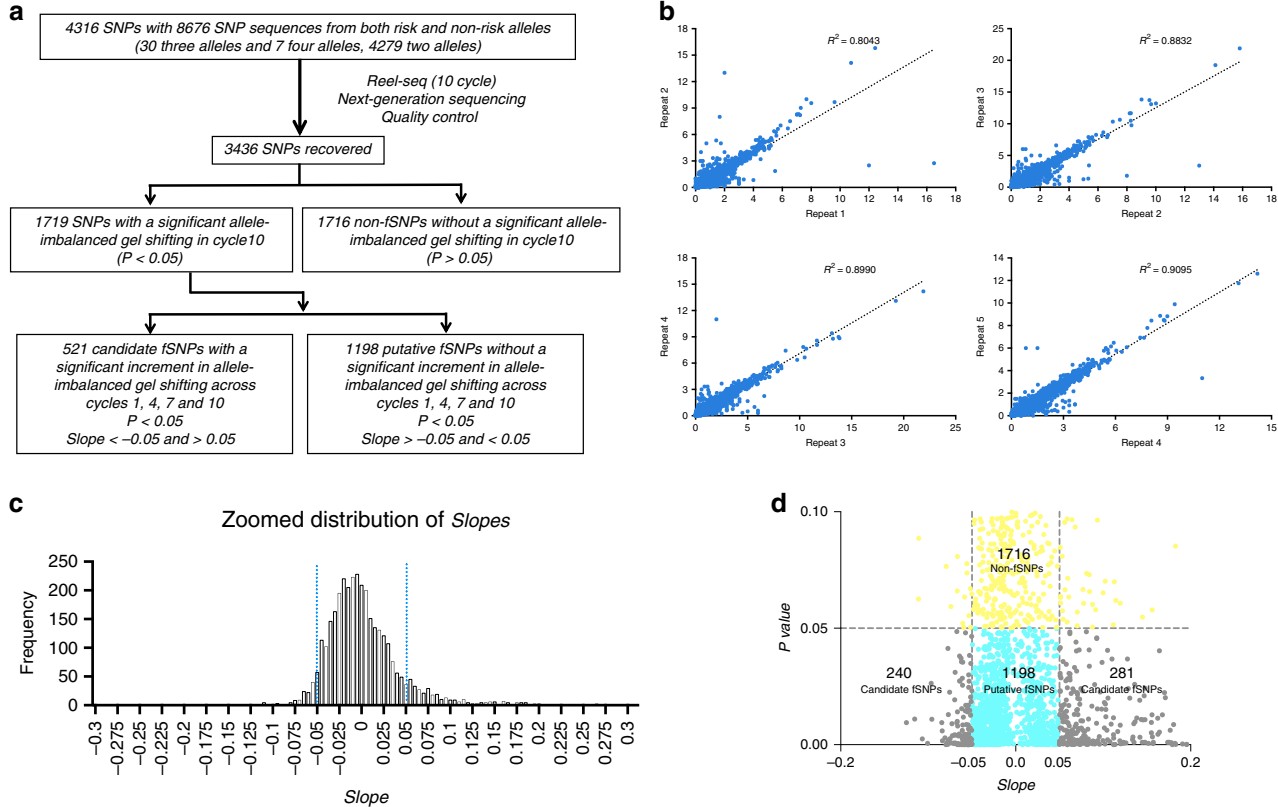

**Fig. 2 Reel-seq screen on 4316 BC-associated SNPs on 177 loci. a** Outline of the Reel-seq screen. **b** Correlation between technical replicates. The ratios of the raw data from five samples versus five controls at cycle 10 were plotted along the x and y axis with $R^2 = 0.80$ for Repeat1/2; $R^2 = 0.88$ for Repeat2/3; $R^2 = 0.89$ for Repeat3/4; $R^2 = 0.90$ for Repeat4/5. **c** Histogram of *Slopes* in Reel-seq data analysis showing an empirical cutoff line (Blue) with $-0.05 > Slope > 0.05$ (zoomed distribution of *Slopes* between $-0.3$ and $0.3$ in *Slopes*). **d** Dot plot of *P* value and *Slope* of 3436 SNPs to identify fSNPs. Candidate fSNPs (gray) with *P* value < 0.05 and $-0.05 > Slope > 0.05$; Putative fSNPs (light blue) with *P* value < 0.05 and $-0.05 < Slope < 0.05$; and non-fSNP (yellow) with *P* value > 0.05. The data was drawn to fit the figure. The full-size plot is in Supplementary Fig. 1. *P* value for HTP Reel-seq screen was calculated with five technical replicates using Student's *t* test with two tails without correction for multiple hypothesis testing. Source data are available in the Source Data file.

corresponding unlabeled probes (Fig. 3a) (Source data are provided as a Source Data file). We also showed the predicted allele-imbalanced gel shifting with the direction between the two alleles consistent with the data obtained from Reel-seq screen. Indeed, all these predicted allele-imbalanced gel shifting are in the same direction with the actual gel shift assays except for rs4752570 (Fig. 3a). To further validate these 12 fSNPs, luciferase reporter assays were performed to assess the allele-imbalanced luciferase activities in MDA-MB-468 cells. Our results revealed significant allelic differences in luciferase activity for all these 12 identified fSNPs (Fig. 3b) (Source data are provided as a Source Data file). Together, these data demonstrate the fidelity of using Reel-seq to identify fSNPs.

**Identification of fSNPs on the BC-associated *FGFR2* locus.** In total, there are 30 SNPs in four LDs with $R^2 > 0.8$ on the BC-associated *FGFR2* locus. All these SNPs are located within intron 2 of the *FGFR2* gene. Besides the five candidate fSNPs, there are 13 putative fSNPs, and 10 non-fSNPs based on our Reel-seq screen (2 *FGFR2* SNPs failed our quality control steps). To further demonstrate the fidelity of Reel-seq, we first performed an allele-imbalanced EMSA with the 13 putative fSNPs, together with the 2 SNPs that did not pass the quality control filter. We identified that in this pool of 15 SNPs, 13 SNPs showing an allele-imbalanced gel shift pattern (Fig. 3c) (Source data are provided as a Source Data file), suggesting a marked enrichment of fSNPs among this pool of putative fSNPs. We next performed another

EMSA on ten non-fSNPs and found that seven out of ten showed no allele-imbalance gel shift activity (Fig. 3d) (Source data are provided as a Source Data file). Together, these data validate both the feasibility and fidelity of Reel-seq to rapidly identify disease-associated fSNPs with a true positive recovery rate of roughly 92% (combination of both candidate and putative fSNPs), and a false negative rate ~30%.

We also compared our Reel-seq findings on the *FGFR2* locus against an in silico analysis using HaploReg 4.1, a web-based tool for epigenetic and functional annotations of genetic variants[23]. We scored each of the 30 *FGFR2* SNPs analyzed in Fig. 3 on a scale of $0-5$ to reflect the number of positive annotations for histone methylation (two markers), DNase hypersensitivity as well as for predicted protein binding and alteration in binding motifs. The average score for the 5 candidate fSNPs, 15 putative fSNPs and 10 non-fSNPs was 3.6, 3, and 3.2, respectively (Supplementary Table 1), indicating that there was no concordance between these two approaches in their capacity to identify fSNPs.

**SDCP-MS to identify proteins specifically binding to fSNPs.** To further understand how BC-associated fSNPs modulate risk gene expression, we developed SDCP-MS to identify fSNP-bound proteins (Fig. 4a). SNP-specific DNA competition pulldown enables us to use a fSNP sequence identified from our Reel-seq as a "bait" to pull down specific regulatory proteins with minimal background. This is achieved by three modifications from

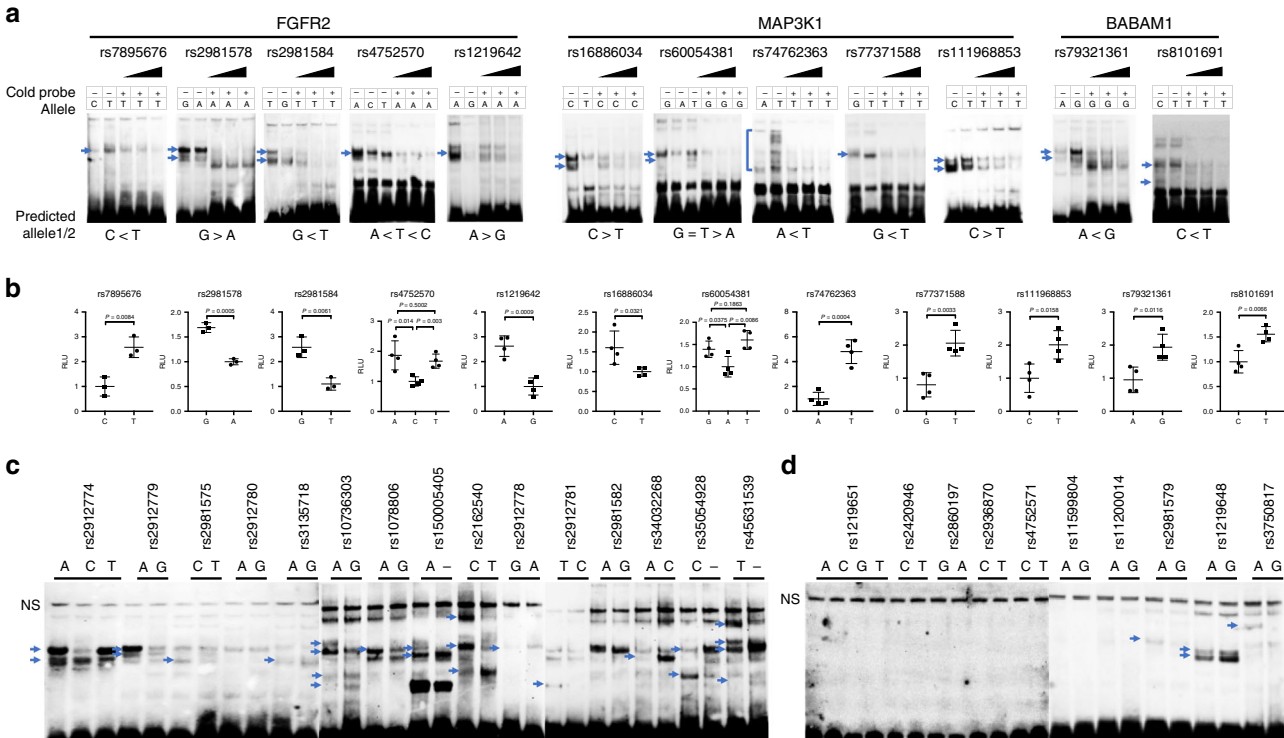

**Fig. 3 Demonstration of fSNPs by EMSA and luciferase reporter assay. a** EMSA showing allele-imbalanced gel shifting for five candidate fSNPs on the *FGFR2* locus (left), five on the *MAP3K1* locus (middle) and two on the *BABAM1* locus (right) with *P* value < 0.05; −0.05 > *Slope* > 0.05. Also, same EMSA showing the allele-imbalanced shifted bands are competed away by cold probe with a concentration of 2, 4 and 6 μg in each assay. Predicted allele1/2: predicted allele-imbalanced gel shifting with the direction between the two alleles based on the data from Reel-seq screen. The data represent three biological independent experiments (*n* = 3). **b** Luciferase reporter assays showing allele-imbalanced luciferase activity for each SNP. The data for rs7895676, rs2981578, and rs2981584 represent three biological independent samples (*n* = 3); the data for the other nine SNPs represent four biological independent samples (*n* = 4). *P* value was calculated using Student's *t* test with two tails. Error bars represent the median with SE. **c** EMSA showing allele-imbalanced gel shift results for 13 of the 15 putative fSNPs (*P* value < 0.05) on the *FGFR2* locus. The data represent three biological experiments (*n* = 3). **d** EMSA showing allele-imbalanced gel shift pattern on three of the ten non-fSNPs (*P* value > 0.05) on the *FGFR2* locus. The data represent three biological experiments (*n* = 3). *P* value was calculated using Student's *t* test with two tails. Error bars in all the plots represent the median with SE. NS nonspecific gel shift, RLU relative luminescence unit. Blue arrows indicating the allele-imbalanced gel shift pattern. Source data are available in the Source Data file.

traditional DNA pulldown assays. First, we build a 5′ biotinylated SDCP construct for each fSNP by flanking a 31 bp fSNP-centered sequence with two restriction enzyme sites, *Eco*RI distally and *Bam*HI proximally. As such, this 31 bp fSNP sequence can be released together with its specific binding proteins from the bead-conjugated SDCP construct with sequential restriction enzyme cutting (e.g. *Eco*RI and *Bam*HI). The proteins that nonspecifically bind to 3′ and 5′ double-stranded DNAs, to single-stranded DNAs, and to the beads can therefore be largely removed. As such, only the proteins released with the fSNP sequence are collected for complex identification by MS analysis. Second, we employ a unique negative competitor that has the same 31 bp fSNP sequence, but with a 3 bp deletion on either side of the SNP site, including the SNP. We refer to this reagent as the 7d competitor. By adding an excessive amount of this negative competitor in the bead-DNA-NE incubation, we can further reduce nonspecific protein binding. Third, we perform the pulldown reaction with multiple different fSNP sequences in parallel so that we can use each fSNP sequence as a negative control. Finally, we performed the pulldown assay with each fSNP in duplicate. By doing these four steps, we could greatly improve the specificity and fidelity of the assay.

We applied SDCP-MS to the three validated fSNPs rs7895676, rs2981578, and rs2981584 on the *FGFR2* locus. Briefly, 15 μg of the purified SDCP construct DNA for each SNP was incubated with 1 mg NE in a buffer containing 600 μg of its 7d competitor.

After a 2 h incubation, the protein-bound DNA-beads were washed and digested with *Eco*RI to remove the 3′ end of DNA and a host of nonspecific proteins. After washing, the protein-bound DNA-beads were further digested with *Bam*HI to release the 31 bp fSNP DNA together with its binding proteins for subsequent MS analysis. In total, we recovered 400 proteins from MS analysis. In all, 283 were filtered for their nonspecific binding to all these three SNPs and 98 for binding to SNPs without reproducibility. By removing these 381 proteins, we identified a total of 19 unique binding proteins that bound to these three fSNPs showed by peptide spectrum counts (Table 1) (Source data are provided as a Source Data file). Eight of these proteins were specific to one fSNP including PARP-2, SERPINC1 and TFAM to rs7895676, and MYH9, TEAD3, EMG1, YBX3, and C1QBP to rs2981578. Ten proteins were shared by two fSNPs including EFTUD2, EXOSC1, HIST1H2AB, RANBP2, TEAD1, NFIB, SNRNP40, MED24, LAD1, and C2orf49. While most of these proteins are reported as transcriptional regulators, TFAM is believed to be a transcription factor that only transcribes mitochondrial DNA. Based on the specificity and perceived functional relevance, we chose PARP-2 and TFAM on rs7895676, TEAD1 and TEAD3 on rs2981578, and NFIB on rs2981584 for further functional analysis. PARP-2 belongs to PARP family of enzyme capable of catalyzing a poly(ADP-ribosyl)ation reaction. It contains a catalytic domain at its C-terminal with no N-terminal DNA binding domain such as PARP-1[24]. PARP-2 was

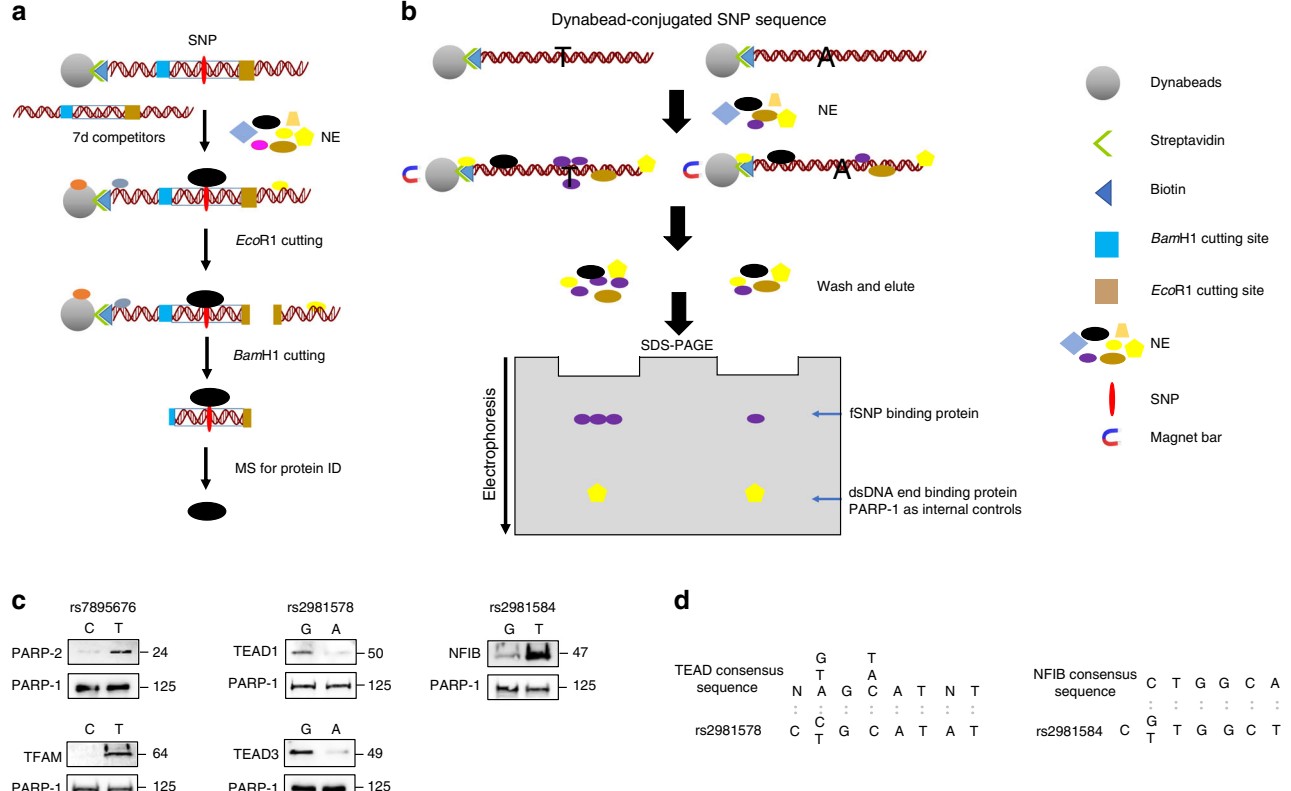

**Fig. 4 SDCP-MS and AIDP-Wb. a** Principle of SDCP-MS. **b** Principle of AIDP-Wb. **c** AIDP-Wb demonstrating the allele-imbalanced binding of PARP-2 and TFAM to rs7895676 (left), TEAD1 and TEAD3 to rs2981578 (middle), and NFIB to rs2981584 (right). The data represent three biological independent experiments (*n* = 3). For rs2981578, G: risk allele, A: non-risk allele; for rs2981584, G: risk allele, T: non-risk allele; and for rs7895676, C: risk allele, T: non-risk allele. PARP-1: a double-stranded DNA end binding protein as internal loading controls for AIDP-Wb showing equal loading of both risk and non-risk allele. **d** Sequence comparison between TEAD consensus sequence and rs2981578 and between NFIB consensus sequence and rs2981584. NE nuclear extract, MS mass spectrometry, ID identification. Source data are available in the Source Data file.

**Table 1 Peptide spectrum counts showing proteins identified by SDCP-MS bind to three fSNPs on the *FGFR2* locus.**

| Gene name | rs7895676-1 | rs7895676-2 | rs2981578-1 | rs2981578-2 | rs2981584-1 | rs2981584-2 |
|---|---|---|---|---|---|---|
| *PARP-2* | 1 | 2 | | | | |
| *EFTUD2* | 9 | 7 | | | 3 | 6 |
| *MYH9* | | | 1 | 1 | | |
| *TEAD3* | | | 10 | 7 | | |
| *EXOSC1* | 3 | 1 | 1 | 1 | | |
| *HIST1H2AB* | 1 | 2 | | | 3 | 2 |
| *RANBP2* | 1 | 3 | 2 | 2 | | |
| *TEAD1* | | | 5 | 5 | 1 | 1 |
| *NFIB* | 3 | 1 | | | 8 | 4 |
| *SNRNP40* | 1 | 3 | | | 3 | 2 |
| *SERPINC1* | 1 | 1 | | | | |
| *TCERG1* | 3 | 3 | 1 | 1 | | |
| *EMG1* | | | 1 | 1 | | |
| *YBX3* | | | 6 | 4 | | |
| *MED24* | 3 | 3 | | | 2 | 2 |
| *C1QBP* | | | 1 | 1 | | |
| *TFAM* | 1 | 1 | | | | |
| *LAD1* | 1 | 2 | 2 | 1 | | |
| *C2orf49* | 2 | 1 | | | 1 | 1 |

Each SNP was performed in duplicate.

reported to negatively regulate SIRT1 expression[25]. Both TEAD1 and TEAD3 are members of the TEAD family of transcription factors and recent studies have highlighted that TEADs, together with their coactivators, promote the development of various malignancies[26,27]. The *NFIB* gene is a part of the *NFI* gene complex that includes *NFIA*, *NFIC*, and *NFIX*[28], which plays important roles in mammary gland development through regulation of key mammary gland-specific genes[29].

**AIDP-Wb verifies a specific protein:fSNP binding**. To quickly demonstrate that PARP-2 and TFAM, TEAD1 and TEAD3, and NFIB specifically bind to their corresponding fSNPs, we developed the AIDP-Wb. This technique was modified from a previous publication[30], and allows for the simultaneous validation of both the fSNP and the fSNP-bound regulatory proteins by Western blot analysis (Fig. 4b). In brief, we used the same two fSNP-centered DNAs representing both the risk and non-risk allele of a fSNP used for allele-imbalanced gel shift assays (Fig. 3). For each fSNP, we attached the same amount of these two alleles to the equivalent amount of streptavidin-coated Dynabeads (Thermo Fisher Scientific). After the DNA-beads were incubated with nuclear proteins isolated from MDA-MB-468 cells, nuclear proteins that bind to the DNAs were magnetically purified and analyzed by Western blot for the two alleles in parallel. The known protein that specifically bind to the fSNP was detected with an antibody against this protein to check if there was any differential binding to the two alleles of the fSNP. To ensure that the same amount of DNA was used for both alleles, we probed the same blot simultaneously with an antibody against PARP-1, a ubiquitous and abundant nonspecific dsDNA end-binding protein that served as an internal loading control. Using this method, we observed that both PARP-2 and TFAM could bind to rs7895676 with the risk allele C having less protein binding (Fig. 4c, left); both TEAD1 and TEAD3 bind to rs2981578 with the risk allele G having more protein binding (Fig. 4c, middle); and NFIB binds to rs2981584 with the risk allele G having less protein binding (Fig. 4c, right) (Source data are provided as a Source Data file). These data support that these identified proteins bind to the three fSNPs on the *FGFR2* locus. In addition, the noted allele-imbalanced bindings of these regulatory proteins verified by the AIDP-Wb further validate that rs7895676, rs2981578, and rs2981584 are indeed bona fide fSNPs.

To further demonstrate the specific binding of these proteins to their corresponding fSNPs, we also investigated the binding motif for these five proteins. We discovered the TEAD family binding sequence with the core sequence 5′-N[A/T/G]G[T/A/C]ATNT-3′[31], which matches the sequence around the rs2981578 5′-C[C/A/T] GCATAT and the NFIB binding motif c/tTGGCa/t[32], which exits in rs2981584 sequence TC[G/T]TGGCTT as shown in Fig. 4d. These findings confirm our data from the AIDP-Wb assay that TEAD1 and TEAD3 are likely rs2981578 binding proteins and NFIB is a rs2981584 binding protein. TFAM regulates both mitochondrial transcription initiation and mitochondrial DNA (mtDNA) copy number by binding to mtDNA with and without sequence specificity[33]. There are two binding motifs identified on the two major promoters of mtDNA: the light strand promoter (LSP) and the heavy strand promoter 1 (HSP1); however, none of these sequences match the sequence on rs7895676. For PARP-2, there is no report found to describe the binding consensus sequences.

**Regulation of FGFR2 expression by fSNP-bound proteins**. To further validate a functional role of the identified regulatory proteins on FGFR2 expression, we performed shRNA knockdown on these five genes in MDA-MB-468 cells and assessed the

subsequent effects on FGFR2 expression. Using shRNA lentiviruses, we were able to generate polyclonal cell lines that show reduced expression of PARP-2 and TFAM (Fig. 5a, left), TEAD1 and TEAD3 (Fig. 5a, middle), and NFIB (Fig. 5a, right) at both the mRNA (Fig. 5a, upper) and protein level (Fig. 5a, lower). Knocking down TFAM, TEAD1, TEAD3, and NFIB resulted in a significant upregulation of FGFR2 (Fig. 5b, left, middle and right), indicating that these proteins likely function as important repressors of FGFR2 expression. In contrast, knockdown of PARP-2 resulted in a significant decreased expression of FGFR2 (Fig. 5b, left), indicating PARP-2 likely functions to activate FGFR2 expression. Regulation of FGFR2 was also evident at both the mRNA (Fig. 5b, upper) and protein level (Fig. 5b, lower), consistent with a defined transcriptional regulatory function for these five SDCP-identified gene products (Source data are provided as a Source Data file).

As mentioned above, *TFAM* gene, located on chromosome 10, encodes mitochondrial transcription factor A that is a key activator of mitochondrial transcription[34]. We are aware of no report demonstrating that TFAM is a transcription factor regulating nuclear DNA expression although it has been observed that nuclear TFAM negatively regulates itself without modulating mitochondrial activity in HT22 hippocampal neuronal cells[35]. To further demonstrate that TFAM is a nuclear transcription factor regulating FGFR2 expression, we performed two additional experiments. First, we overexpressed TFAM in MDA-MB-468 cells and we observed a decreased expression of FGFR2 at both the protein and mRNA level (Fig. 5c). Second, we performed a luciferase reporter assay with a construct that contains the T allele from rs7895676 as previously used in Fig. 3b. Our data demonstrated that in MDA-MB-468 cells, increased expression of TFAM or siRNA-mediated decreased TFAM expression resulted in a corresponding decrease and increase in luciferase activity (Fig. 5d). As a control, we also performed the same experiment with a luciferase reporter construct containing an irrelevant SNP sequence. In this case, we did not observe any difference between the samples and controls (Fig. 5d) (Source data are provided as a Source Data file). Together, these data validate that TFAM can act to regulate nuclear transcription, and in particular, this factor can negatively affect FGFR2 expression in MDA-MB-468 cells. These observations are also consistent with our shRNA knockdown data as shown in Fig. 5a.

Finally, since most of the mutations in the *FGFR2* gene were identified in ER+ BC[20], not in TNBC represented by MDA-MB-468 cells, we also investigated the regulation of FGFR2 expression by these five proteins in the ER+ BC cell line MCF7. As we observed in MDA-MB-468 cells, all factors regulated FGFR2 although the directionality of this regulation often differed between MCF7 and MDA-MB-468 cells (Fig. 5e) (Source data are provided as a Source Data file). These data, while at present incompletely understood, are consistent with the observations that the expression of *FGFR2* in various cancer cells can be either increased or decreased[20].

## Discussion

In an effort to identify disease-associated fSNPs, we developed Reel-seq, an in vitro technique requiring no more challenge than EMSA and PCR. Previously, we developed SNP-seq, a type IIS restriction enzyme protection-based assay to identify fSNP[12]. However, SNP-seq depends on type IIS restriction enzyme binding; therefore, SNP-seq constructs with mutations in the type IIS restriction enzyme binding sites introduced by massive parallel oligonucleotide synthesis and PCR amplification will be positively selected. Even though these mutated SNP sequences will be eliminated by sequence quality control, this will reduce the

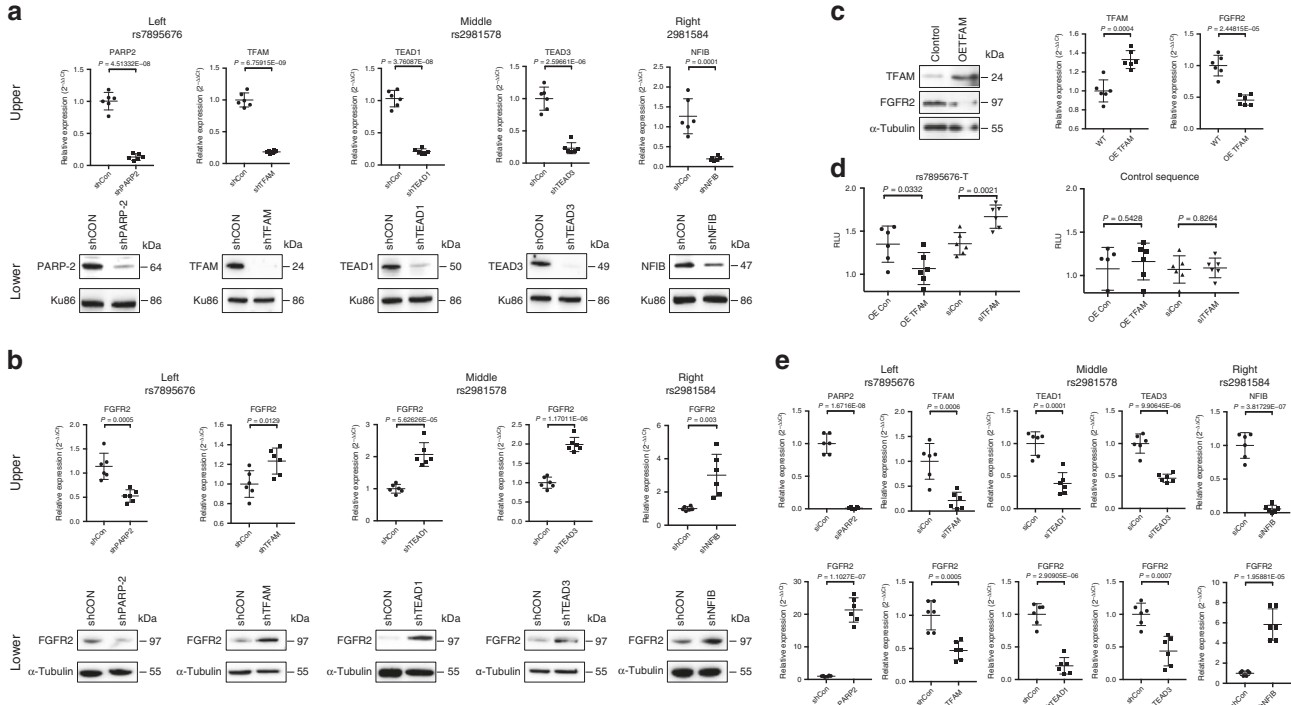

**Fig. 5 Regulation of *FGFR2* transcription by PARP-2 and TFAM, TEAD1 and TEAD3, and NFIB. a** shRNA (sh) knockdown of PARP-2 and TFAM, TEAD1 and TEAD3, and NFIB, in MDA-MB-468 cells. Upper: result from qPCR ($n = 3$ biological independent experiments with each in duplicate); Lower: results from Western blot ($n = 3$ biological independent experiments). Ku86: employed as a loading control for nuclear extracts. **b** Modulation of *FGFR2* expression in the MDA-MB-468 cells following knockdown of PARP-2 and TFAM, TEAD1 and TEAD3, and NFIB. Upper: results from qPCR ($n = 3$ biological independent experiments with each in duplicate); Lower: results from Western blots ($n = 3$ biological independent experiments). α-Tubulin: a loading control for cytosolic proteins. **c** Overexpression of TFAM downregulates FGFR2 expression as shown by Western blot (left) ($n = 3$ biological independent experiments) and qPCR (right) ($n = 3$ biological independent experiments with each in duplicate). OE overexpression. **d** Luciferase reporter assays demonstrating TFAM regulates luciferase activities through rs7895676(T) evidenced by either overexpression (OE) or siRNA (si) knockdown in MDA-MB-468 cells. The data represent six biological independent samples ($n = 6$). **e** FGFR2 expression following siRNA (si) knockdown of PARP-2 and TFAM (left), TEAD1 and TEAD3 (middle), and NFIB (right) in MCF7 cells. The data from Western blots represent three biological independent experiments ($n = 3$) and the data from qPCR represent three biological independent experiments with each in duplicate ($n = 3$). $P$ value was calculated using Student's $t$ test with two tails. Error bars in all the plots represent the median with SE. Source data are available in the Source Data file.

NGS reads, thereby complicating the data analysis[12]. To overcome this limitation, in this study, we developed Reel-seq (Fig. 1), a new type of in vitro unbiased HTP technique to identify fSNPs. Since Reel-seq detects DNA:protein binding directly based on EMSA, it circumvents the mutations introduced by SNP-seq. However, both Reel-seq and SNP-seq are techniques based on in vitro assays. As such, these strategies cannot uncover fSNPs that effect gene expression through splicing or regulatory RNAs, or rely on the 3D structure of the DNA in addition to their recognition motifs.

Similar to Reel-seq, SNPs-seq was recently developed by also detecting allele-specific binding between proteins and SNPs[36]. However, it uses a spin column to identify fSNPs by analyzing protein bound to ds-oligos, instead of the free oligos in the flow through[36]. Therefore, SNPs-seq detects the allele-imbalanced binding of a fSNP to its cognate binding proteins with only one round of enrichment, which could ultimately affect the sensitivity of this assay for identifying fSNPs. Since Reel-seq collects the unshifted library (equivalent to the free oligos in the flow through) on each cycle of allele-imbalanced gel shifting, this greatly increases the sensitivity of this method. Indeed, out of 3436 BC-associated SNPs, using Reel-seq we identified 1719 SNPs with significant allele-imbalanced gel shifting ($P$ value < 0.05). This is a recovery rate of 50%. In contrast, using a pool of SNPs that were enriched for fSNPs by functional annotations, the rate of recovery for SNPs-seq was only 27% (101 out of 374)[36].

Nonetheless, in the future, it will be important to directly compare these two screening techniques using the same SNP library.

In addition, as we present in Supplementary Table 1, we compared our screening results on the *FGFR2* locus with that obtained by functional annotations from Haploreg 4.1. We did not observe any obvious correlation between these two approaches. The same conclusions were reached with several other Reel-seq screens carried out in our lab (unpublished data). Based on these observations, currently we do not believe that it is useful to pre-screen disease-associated SNPs with functional annotations before using Reel-seq.

We previously described FREP-MS as a method to identify regulatory proteins that potentially bind to fSNPs[12]. Here, we have described SDCP-MS, which uses excess 7d competitor during the incubation, and thereby greatly increases the specificity and fidelity of FREP-MS to identify fSNP-bound proteins. In addition, as we described above, SDCP-MS is fundamentally different from other DNA pulldown assays, as well as antibody pulldown assays. Instead of eluting proteins from DNA conjugated beads that have been pulled down, it collects proteins specifically by releasing the SNP sequence from the pulldown using a sequential restriction enzyme digestion. By doing so, it eliminates proteins that bind to 3′ and 5′ dsDNA, to single-stranded DNA as well as to the beads. This, together with the 7d competitor, reduces the nonspecific protein burden.

Conventionally, validation of the specific binding of a regulatory protein to a fSNP can be carried out by gel supershift assay and ChIP assay for protein:DNA binding. However, both gel supershifting and ChIP assays generally require antibodies with extremely high quality and specificity. To overcome this difficulty, we developed the AIDP-Wb that allows us to detect specific protein:fSNP binding, instead of protein:DNA binding, using conventional Western blot analysis. Moreover, since AIDP-Wb detects the binding of a regulatory protein to a particular fSNP, we can validate both the regulatory protein and its binding to the fSNP simultaneously.

Using Reel-seq with NE isolated from MDA-MB-468, we validated five candidate fSNPs ($P$ value < 0.05 and a *Slope* < −0.05 and >0.05) on the BC-associated *FGFR2* locus. In 2008 and 2013, Meyer et al. first dissected the BC-associated *FGFR2* locus and identified rs2981578, together with rs7895676, rs35054928, and rs45631563 as candidate fSNPs by using functional annotation and gel shift assay[37,38]. Of note, both rs2981578 and rs7895676 were identified by Reel-seq as candidate fSNPs, while we identified rs35054928 as a putative fSNP (Fig. 3a–c). In contrast, rs45631563, which is on another LD on the *FGFR2* locus, was not included in our Reel-seq library. Based on the motif analysis, OCT-1 and RUNX2 were previously identified as proteins that specifically bound to rs2981578, which was confirmed by gel supershifting[37,38]. FOXA1 was shown to bind to the same fSNP by ChIP assay[38]. To understand the relevance of these findings to our analysis, we first performed AIDP-Wb with these three proteins using NE isolated from MDA-MB-468 cells. Our results showed that OCT-1 did not bind to rs2981578, while both RUNX2 and FOXA1 could bind to this fSNP, with only RUNX2 showing evidence for allele-imbalanced binding, albeit weakly (Supplementary Fig. 2a). To further check if these factors modulated FGFR2 expression, we knocked down all these three proteins in MDA-MB-468 cells. We observed decreased expression of FGFR2 in RUNX2 knockdown cells (Supplementary Fig. 2b, c, middle). Surprisingly, we also observed decreased expression of FGFR2 in OCT-1 knockdown cells while the expression of FGFR2 was unchanged in FOXA1 knockdown cells (Supplementary Fig. 2b, c, upper and lower). Based on these data, we believe that OCT-1 is an FGFR2 regulator, but that it does not act via binding to rs298578, which is the likely reason that it was not identified in our SDCP-MS assay (Supplementary Fig. 2a, upper). In contrast, RUNX2 likely regulates FGFR2 expression via its binding to rs2981578. However, due to its very low binding affinity to rs2981578 (Supplementary Fig. 2a, middle), it was not identified by our SDCP-MS, highlighting a limitation in the sensitivity of the current technique. The equivalent binding of FOXA1 to both alleles of rs298578, and the absence of an effect on FGFR2 expression following knockdown of FOXA1 (Supplementary Fig. 2, lower), suggests that this factor may not possess SNP-specific binding (Source data are provided as a Source Data file). Nonetheless, further investigation of the regulation of FGFR2 by these proteins in different BC cell lines is ultimately needed to further validate these observations.

Collectively, our results demonstrate that sequential application of Reel-seq, SDCP-MS, as well as the AIDP-Wb, enables us to identify and characterize disease-associated fSNPs and fSNP-bound proteins in a highly efficient manner. In addition, while our results begin to reveal the mechanisms underlying the contribution of BC-associated fSNPs, they also highlight the complexity of gene transcriptional regulation in BC development.

## Methods

**Cells and culture.** Human BC cells MDA-MB-468 (Cat#: ATCC HTB-132) and MCF7 (Cat#: ATCC HTB-22) were purchased from ATCC and were free of mycoplasma. MDA-MB-468 cells were cultured in F12:DMEM (1:1) medium and MCF7 cells in RPMI 1640 medium. Both were supplemented with 10% fetal bovine serum.

**Primers and antibodies.** All primers are listed in Supplementary Table 2 and were purchased from IDT. All antibodies are listed in Supplementary Table 3 with the vendor's information.

**Reel-seq.** The Reel-seq construct for generating the BC library was the same as the SNP-seq construct described previously[12]. Primer bioseq and G3 were used for library amplification and regeneration during the screen. For screening, ~10 µg NE (buffer for control) was mixed with ~50 ng library DNA in the binding buffer from the LightShift™ Chemiluminescent EMSA Kit (Thermo Fisher Scientific) and incubated at RT for 2 h. The reaction was performed with five buffer-treated controls and five NE-treated samples. These samples were then run on a 6% TBE native gel for gel shifting. After the completion of electrophoresis, unshifted bands at 75 bp from each of the controls and samples were cut and isolated. The isolated library DNA was amplified by PCR with bioseq and G3 primers using Accuprime Taq polymerase (Invitrogen) and the regenerated libraries were used for the next cycle of the Reel-seq screen. A total of ten cycles were performed. After the screen, NGS was performed by NextSeq 500 system with the PCR product from cycles 1, 4, 7 and 10 analyzed.

For quality control, we eliminated SNPs for which complete sequencing data were not available across cycles 1, 4, 7 and 10. To identify fSNPs, we first calculated the ratio of the sequence count between the two alleles for each SNP, and for each of the five replicates at cycle 10. We identified SNPs demonstrating allele-imbalanced gel shifting at cycle 10 with a significant difference in the ratio between the five controls and the five samples using a Student's $t$ test with a $P$ value < 0.05. Second, we identified SNPs for which allele-imbalanced gel shift differences increased with increasing cycle number, indicating progressive enrichment. For each SNP, we calculated the average ratio of the sequence count between alleles from the five buffer-treated controls and the five NE-treated samples and normalized the ratio of the sample with the control. We next used the normalized ratio from cycles 1, 4, 7 and 10 to calculate for *Slope* and identified SNPs with a *Slope* > 0.05 and < −0.05 as an empirical cutoff point. In this way, we identified candidate fSNPs with a $P$ value < 0.05 and a *Slope* > 0.05 and < −0.05 and putative fSNP with a $P$ value < 0.05 and a *Slope* < 0.05 and > −0.05.

**Electrophoretic mobility shift assay (EMSA).** Electrophoretic mobility shift assay was performed using the LightShift Chemiluminescent EMSA Kit (Thermo Fisher) according to the manufacturer's instructions. For the probe, a 31 bp SNP fragment with the SNP centered in the middle was made by annealing two oligos. The double-stranded oligos were then biotinylated using the Biotin 3′ End DNA Labeling Kit (Thermo Fisher Scientific). Nuclear extract from MDA-MB-468 was isolated using NE-PER Nuclear and Cytoplasmic Extraction Reagents (Thermo Scientific) according to the manufacturer's instructions. For competition assay in EMSA, 2, 4, and 6 µg cold competitors were added together with NE. The data represent three biological repeats.

**SDCP-MS.** SNP-specific DNA competition pulldown-mass spectrometry was modified from our previously described FREP-MS[12] by the addition of 40-fold excess of a 7d competitor, a negative competitor that has the same fSNP sequence, but contains a 3 bp deletion on both sides of the SNP site, including the SNP. In brief, ~15 µg of the purified SDCP construct DNA was conjugated to 150 µl streptavidin-coupled Dynabeads (Life Technologies) according to the manufacturer's instruction. The DNA-beads were then washed and mixed with 1 mg NE in a buffer containing 40-fold excess of 7d competitor at RT for 1 h. After separation and washing, the protein-DNA-beads were digested with 5 µl *Eco*RI (100 units/µl NEB) at 37 °C for 30 min to remove the 3′ DNA plus the proteins that bound to this non-SNP region. After separation and washing, the protein-DNA-beads were subsequently digested with 5 µl *Bam*HI (100 units/µl NEB) at 37 °C for 45 min to release the fSNP sequence plus the fSNP-bound proteins. The supernatant was run on an 8% short SDS-PAGE gel (http://www.bidmcmassspec.org/) and then collected for protein complex identification by mass spectrometry. For mass spectrometry analysis, two technical replicates were performed in parallel for each SNP.

To identify fSNP-specific binding proteins, we first eliminated all the proteins that have peptide counts in all the samples. We also eliminated all the proteins that had peptide spectrum counts only in one of the two replicates for each fSNP. fSNP-specific bound proteins were identified as those proteins that have protein peptide spectrum counts in both of the two replicates for one fSNP, but, not in the two replicates for the controls performed in parallel.

**AIDP-Wb.** Nuclear extract was isolated as described above. A 31 bp biotinylated SNP sequence centered with either the risk or non-risk allele was generated by annealing biotinylated primers purchased from IDT. Approximately 1 µg DNA was attached to Dynabeads™ M-280 Streptavidin. DNA-beads were mixed with ~100 µg NE at RT for 1 h with rotation. After separation and washing, the DNA-bound proteins were eluted for Western blot analysis. The data represent three biological replicates.

**Western blot**. Whole-cell proteins were isolated with RIPA buffer (Sigma). Cytosolic proteins and nuclear proteins were isolated using NE-PER Nuclear and Cytoplasmic Extraction Reagents (Thermo Fisher Scientific) according to the manufacturer's instructions. The data represent three biological replicates.

**qPCR**. Total RNA was isolated using RNeasy Mini Kit (Qiagen). cDNA was synthesized with SuperScript® III Reverse Transcriptase (Invitrogen) after the RNA sample was treated with DNase I (Invitrogen). All procedures were performed in accordance with the manufacturer's protocols. qPCR was done with the StepOne real-time PCR system according to the protocol provided for the power SYBR green PCR master mix (Applied Biosystems). All the data represent a combination of three biological replicates unless as indicated.

**shRNA and siRNA knockdown**. For shRNA stable knockdown in human MDA-MB-468 cells, lentiviruses were obtained from a MISSION® shRNA Library (Sigma) (Supplementary Table 3) and the knockdown was performed according to the manufacturer's protocol. For siRNA transient knockdown in human MDA-MB-468 and MCF7 cells, siRNAs were purchased from Horizon Discovery and the knockdown was performed according to the manufacturer's protocol. Additional information regarding shRNAs and siRNAs are provided in Supplementary Table 3.

**Luciferase report assay**. Luciferase reporter assays were performed by pGL3 Luciferase Reporter Vectors (Cat#:E1751, Promega). The luciferase activity was measured by the Dual-Glo® Luciferase Reporter Assay System (cat#: E2920, Promega). All experiments were performed according to the manufacturer's protocol. Insert target sequences are listed in Supplementary Table 4. The data represent biological independent samples as indicated.

**Statistical analysis**. $P$ value was calculated using Student's $t$ test with two tails without correction for multiple hypothesis testing. Error bars represent the median with SE. For EMSA and Western blot, the data in each case represent three biological independent experiments. For qPCR, the data represent a combination of three biological independent samples. For luciferase reporter assay, the data represent biological independent samples as indicated.

**Reporting summary**. Further information on research design is available in the Nature Research Reporting Summary linked to this article.

## Data availability

All relevant data are available with the article and supplementary files or in the Source data file. The sequences in the synthetic DNA library used for Reel-seq are available from the authors upon request. Any other relevant information are available upon reasonable request. Source data are provided with this paper.

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

## Acknowledgements

This work was mainly supported by funding granted to G.L. from Aging Institute, UPMC. It was also partly supported by WCRC pilot cancer grant from UPMC Magee-Women's Hospital and grants from National Multiple Sclerosis Society (G.L.), and NIH/NINDS R21 NS096443 (G.L.), NIH/NIAMS R21 AR070378 (G.L.) and NIH/NIA RO1 AG056279A (G.L.).

## Author contributions

G.L. developed Reel-seq, SDCP-MS and AIDP-Wb, designed the study, performed Reel-seq, SDCP and analyzed the data, drafted and revised the manuscript. Y.Z., supported by M.Q. in ECNU and Chinese Scholarship Council (CSC), performed EMSA, luciferase reporter assays, AIDP-Wb, RNAi interference and Western blots in G.L. laboratory and revised the manuscript. T.W., W.S., D.J., and X.Z. assisted with AIDP-Wb, PCR and Western blot. D.W. and J.C. performed sequencing data analysis and statistical analysis. J.Z. provided breast cancer Reel-seq library. S.O. and T.F. assisted with manuscript revision.

## Competing interests

The authors declare no competing interests.
