## [Peer Review File · Nature Communications]

Reviewers' Comments:

Reviewer #1:

Remarks to the Author:

In this paper Zhao et al. present a sequential methodology to identify functional SNPs in GWAS for breast cancer risk loci. This is an area of active research as the functional dissection of risk loci have lagged behind their accelerated pace of discovery. The authors present three sequential methods, Reel-Seq (Regulatory element sequencing), SDCP-MS (SNP-specific DNA competition) and AIDP (allele-imbalance DNA pull down), two of which are variations and improvements on methods previously published by the group. The authors start from a library of 4,316 SNPs associated with breast cancer risk and identify 523 candidate functional SNPs using Reel-seq. The authors then apply SDCP-MS and AIDP to identify multiple regulatory factors that bind regions containing SNPs in an allele-specific manner at the FGFR2 locus.

While there are interesting observations and the methodology seems promising the paper straddles the line between a methods and a discovery paper. Unfortunately, as it stands, the paper falls short of both goals.

As a methods paper it has three important drawbacks: first, there are few methodological details for an independent replication of the methods (for example the mass spectrometry data acquisition, filtering and analysis have very cursory descriptions). Second, there is no attempt to benchmark the method, for example by examining whether the SNPs found agree with SNPs found through in-depth analysis already conducted or with other high-throughput approaches (Nat Commun. 2018 May 22;9(1):2022). Finally, it lacks orthogonal validation, repeating the EMSA in a low through-put fashion is good but is far from validating the method, particularly when more traditional EMSA controls (such as competition) are still needed.

As a discovery paper there is surprisingly little attention to previous work done in general and at those chosen loci. The fact that several proteins identified do not have identified roles as transcription including some that are mitochondrial or are commonly found in mass spectrometry experiments (please check CRAPome database) factors makes these results difficult to believe without further validation. For the FGFR2 locus the authors should comment on how the results can be reconciled with previous work (PLoS Biol. 2008 May 6;6(5):e108.; not cited by the authors) providing evidence for a role of Oct-1/Runx2 as the mediator in this locus.

Minor issues

Rationale for the use of MD –MD-468 cell line. In most studies of cancer risk, the consensus is that normal immortalized cell lines are better models than transformed cell lines. The authors may have technical reasons (ease of culture, high volume, or rapid growth) to use this model but they should a) justify its use, and b) include in the discussion that this is a limitation. There is no mention about mycoplasma testing and Short Tandem Repeat analysis to guarantee cell line identity and quality, in particular, since it is obtained from another laboratory.

Although discovery has lagged behind, there have been several cancer risk loci that have been dissected in depth and the authors barely mention these studies in the intro and discussion.

What is the rationale for using the nominal $p \leq 0.05$ as a significance threshold? If the authors have an argument not to adjust for multiple testing it should be articulated in the manuscript.

It is unclear how the 12 SNPs were picked. Were they the only ones in the set of 523 remaining associated at the chosen loci? According to the authors these were chosen because of their presumed biological relevance since FGFR2 and MAP3K1 were among the earliest loci demonstrating a strong association with breast cancer. Not sure why being the earliest loci found is necessarily a reflection of their biological significance.

Reviewer #2:

Remarks to the Author:

This paper tackles the important problem of identifying, out of a larger LD locus containing many SNPs, the SNP that is causal (disease-causing).

To this end, the authors developed a EMSA-based high throughput method called REEL-seq and applied it to a large set of breast cancer associated SNPs. Several identified SNPs around FGFR2 were further validated, both confirming that proteins differentially bind to probes displaying both alleles, as well as by identifying the binding proteins. Finally, the authors showed that knockdown of these binding proteins indeed affects the expression of FGFR2.

Given that most disease-associated SNPs are located in non-coding regions, I believe that the work described in this manuscript is an important contribution. Hence I recommend acceptance after addressing the following minor issues.

Minor comments:

* Line 152: I was wondering if the P-values were corrected for multiple testing.

* How does the ratio curve look like for the 1197 putative fSNPs with P value < 0.05, but Slope < 0.05 or > -0.05.

* This sentence

The SNPs presumably exert their function by binding to specific regulatory proteins should be replaced by

The regulatory elements containing these SNPs presumably exert their function by binding to specific regulatory proteins

* It would be helpful to indicate in Figure 3 which of the alleles had a different ratio in REEL-seq and whether this allele was detected in the shifted or unshifted fraction. This would allow readers to evaluate whether the effect seen REEL-seq goes into the same direction.

* please define NE on page 6

* please fix some typos: "assocaited" "We apply these technique to"

* Figure 2 has white boxes overlapping the arrowheads

Suggestion:

Maybe the authors could intersect the fSNPs, putative fSNPs and the remaining SNPs deemed to be non-relevant with regulatory elements in the breast cancer cell line (ATAC-seq, TF Chip-Seq data etc). One would expect that the fSNPs (and to a lower extent the putative fSNPs) overlap such regulatory elements more often than the non-relevant SNPs.

University of Pittsburgh

School of Medicine

Aging Institute

Response to reviewer #1

We are happy to see the critical, but rather encouraging comments from our reviewer #1. We have now revised our MS in response to our reviewer#1 's comments.

1). As a method paper it has three important drawbacks: first, there are few methodological details for an independent replication of the methods (for example the mass spectrometry data acquisition, filtering and analysis have very cursory descriptions).

We apologize for the initial lack of details. In this revised version we have extensively re-written our manuscript including adding methodological details on Reel-seq (**line 136 to 151** and **line 525 to 538**), and on SDCP-MS (**line 229 to 260** and **563 to 567**).

2). Second, there is no attempt to benchmark the method, for example by examining whether the SNPs found agree with SNPs found through in-depth analysis already conducted or with other high-throughput approaches (Nat Commun. 2018 May 22;9(1):2022).

In this revised version, we have now benchmarked Reel-seq by comparing its attributes to a number of other strategies including MPRA, PWAS, SNP-seq, and SNPs-seq (Nat Commun. 2018 May 22;9(1):2022). These are analyzed in the Discussion (**line 368 to 417**). While we believe our approach is superior, in the future, a direct head-to-head comparison using the same SNP library would be required for any substantive claim of superiority. We have also evaluated SDCP-MS in comparison with conventional DNA pulldown or antibody (Discussion **line 429 to 440**). Unfortunately, there are very few post-GWAS functional studies using high throughput techniques making it difficult to thoroughly benchmark Reel-seq, SDCP and AIDP, although this underscores the critical need of these new approaches.

University of Pittsburgh

School of Medicine

Aging Institute

3). Finally, it lacks orthogonal validation, repeating the EMSA in a low through-put fashion is good but is far from validating the method, particularly when more traditional EMSA controls (such as competition) are still needed.

We have now validated the 12 candidate fSNPs by repeating the EMSA with cold competitors as shown in **Fig. 3A**. In addition, we have now also validated these 12 candidate fSNPs with allele-imbalanced luciferase reporter assay as shown in **Fig. 3B**.

4). As a discovery paper there is surprisingly little attention to previous work done in general and at those chosen loci.

We apologize for this oversight. In this revised manuscript, we have now extensively reviewed the breast cancer-associated *FGFR2* locus. **a)** We added an entire section **Identification of fSNPs on the breast cancer-associated *FGFR2* locus (line 203 to 216)**, in which, we described the characterization of all thirty SNPs using EMSA in 4 LDs with $R^2 > 0.8$ on the breast cancer-associated *FGFR2* locus. **b)** We compared our results with previously published data on identifying fSNPs on the *FGFR2* locus and found a correlation between these two results (**line 458 to 464**).

5). For the *FGFR2* locus the authors should comment on how the results can be reconciled with previous work (PLoS Biol. 2008 May 6;6(5):e108.; not cited by the authors) providing evidence for a role of Oct-1/Runx2 as the mediator in this locus.

We also performed AIDP-Western blots and RNAi knockdown to verify the regulatory function of OCT-1, RUNX2 and FOXA1 as *FGFR2* transcriptional regulators (**Supplemental figure 2 and line 464 to 477**) and explain the possible reasons for the difference between the published data and our findings (**line 477 to 488**).

6). The fact that several proteins identified do not have identified roles as transcription including some that are mitochondrial or are commonly found in

University of Pittsburgh

School of Medicine

Aging Institute

mass spectrometry experiments (please check CRAPome database) factors makes these results difficult to believe without further validation.

We understand the reviewer's concern about the specificity and fidelity of SDCP, especially about the finding that TFAM is a FGFR2 transcriptional regulator. We have therefore performed two additional experiments to demonstrate that TFAM is a transcriptional regulator modulating FGFR2 expression. **A)** We over-expressed TFAM in MDA-MB-468 cells and showed that increased expression of TFAM results in reduced expression of FGFR2 (**Fig. 5C and line 333 to 341**). **B)** We performed a luciferase reporter assay with a construct containing the T allele from rs7895676 and showed that increased expression of TFAM or siRNA-mediated decreased TFAM expression resulted in a corresponding decrease and increase in luciferase activity (**Fig. 5D and line 341 to 351**). At the same time, we wish to emphasize that our SDCP is not a really DNA pulldown assay, therefore, it is not clear whether the data from the CRAPome database that were generated by DNA or antibody pulldown-MS are as relevant (see Discussion **line 432 to 440**).

7). Rationale for the use of MD –MD-468 cell line. In most studies of cancer risk, the consensus is that normal immortalized cell lines are better models than transformed cell lines. The authors may have technical reasons (ease of culture, high volume, or rapid growth) to use this model but they should a) justify its use, and b) include in the discussion that this is a limitation.

We agree to our reviewer that non-transformed immortalized cell lines are better models than transformed cell lines. The reason that we used MDA-MB-468 cells, instead of immortalized cells, is that we are planning to perform our Reel-seq screen with nuclear extract from different subtypes of breast cancers and to identify subtype-specific fSNPs. MDA-MB-468 cell line is one of the subtypes, the triple-negative/basal-like breast cancer. To overcome the limitation of using this transformed MDA-MB-468 cells with uncharacterized background, we also performed our RNAi knockdown assay in MCF7, a an ER+ breast cancer as shown in **Fig. 5E. (line 353 to 360)**.

University of Pittsburgh

School of Medicine

Aging Institute

8). What is the rationale for using the nominal $p \leq 0.05$ as a significance threshold? If the authors have an argument not to adjust for multiple testing it should be articulated in the manuscript.

We now explain why we used the nominal $P \leq 0.05$ as a significant threshold in **line 166 to 171**. We want to maintain as many potential SNPs as possible; therefore, we didn't use any multiple testing adjustment for the P value. In this case, we are aware of the probability of excessive false positives at the end of our data analysis using only the Reel-seq screen. However, later downstream validation steps, such as allele-imbalanced gel shifting and luciferase reporter assays, were used to narrow this initial pipeline.

9). It is unclear how the 12 SNPs were picked. Were they the only ones in the set of 523 remaining associated at the chosen loci? According to the authors these were chosen because of their presumed biological relevance since FGFR2 and MAP3K1 were among the earliest loci demonstrating a strong association with breast cancer. Not sure why being the earliest loci found is necessarily a reflection of their biological significance.

We now hopefully better explain why we analyzed the 12 candidate fSNPs on FGFR2, MAPK3 and BABAM1 loci **in line 175 to 186**.

Response to reviewer #2

We are also very happy to see the positive comments from our reviewer #2. In addition to the correction on the errors pointed out by this reviewer, we have also performed the following revision:

1). I was wondering if the P-values were corrected for multiple testing.

University of Pittsburgh

School of Medicine

Aging Institute

This was also raised by Reviewer #1. We now explain the reason why we used the nominal $P \leq 0.05$ as a significant threshold in **line 167 to 171** as we also responded to reviewer 1 (point #8 above).

2) How does the ratio curve look like for the 1197 putative fSNPs with P value < 0.05, but Slope < 0.05 or > -0.05.

We now include a dot plot with the ratio of allele1 versus allele2 after we normalized our samples with the controls in the function of *slope* and *P* value as shown in a new **Fig. 2D**;

3) It would be helpful to indicate in Figure 3 which of the alleles had a different ratio in REEL-seq and whether this allele was detected in the shifted or unshifted fraction. This would readers to allow to evaluate whether the effect seen REEL-seq goes into the same direction.

We have now added the predicted allele-imbalanced gel shifting with the direction between the two alleles in **Fig. 3A** to show all are consistent with the actual allele-imbalanced EMSA data except for rs4752570 (**line194 to 197**).

4) Maybe the authors could intersect the fSNPs, putative fSNPs and the remaining SNPs deemed to be non-relevant with regulatory elements in the breast cancer cell line (ATAC-seq, TF Chip-Seq data etc). One would expect that the fSNPs (and to a lower extent the putative fSNPs) overlap such regulatory elements more often than the non-relevant SNPs.

We compared our Reel-seq findings on the *FGFR2* locus against an *in silico* analysis using HaploReg 4.1, a web-based tool for epigenetic and functional annotation of genetic variants as suggested. Unfortunately, we didn't find a consistent overlap between these two approaches in their capacity to identify fSNPs (**Supplemental table 1**).

University of Pittsburgh

School of Medicine

Aging Institute

Reviewers' Comments:

Reviewer #1:

Remarks to the Author:

I am satisfied with the changes.

Genes symbols should be consistently italicized.

Reviewer #2:

Remarks to the Author:

The authors have fully addressed my comments.

There is still a minor issue with Figure 2 having white boxes overlapping the text, but maybe this is a pdf artifact.

Response to reviewer #1:

Reviewer #1 (Remarks to the Author): Genes symbols should be consistently italicized.

We have made the change.

Response to reviewer #2:

Reviewer #2 (Remarks to the Author): There is still a minor issue with Figure 2 having white boxes overlapping the text, but maybe this is a pdf artifact.

We have checked. The reviewer might be right. But we regenerated this Figure 2 in .ai format.